# Alternative Pre-mRNA Splicing of the Mu Opioid Receptor Gene, *OPRM1*: Insight into Complex Mu Opioid Actions

**DOI:** 10.3390/biom11101525

**Published:** 2021-10-15

**Authors:** Shan Liu, Wen-Jia Kang, Anna Abrimian, Jin Xu, Luca Cartegni, Susruta Majumdar, Patrick Hesketh, Alex Bekker, Ying-Xian Pan

**Affiliations:** 1Department of Anesthesiology, Rutgers New Jersey Medical School, Newark, NJ 07103, USA; sl1848@njms.rutgers.edu (S.L.); wj.kang@rutgers.edu (W.-J.K.); aa2279@njms.rutgers.edu (A.A.); jx242@njms.rutgers.edu (J.X.); ph396@newark.rutgers.edu (P.H.); bekkeray@njms.rutgers.edu (A.B.); 2Department of Chemical Biology, Ernest Mario School of Pharmacy Rutgers University, Piscataway, NJ 08854, USA; Luca.cartegni@pharmacy.rutgers.edu; 3Center for Clinical Pharmacology, University of Health Sciences & Pharmacy and Washington University School of Medicine, St. Louis, MO 63110, USA; susrutam@email.wustl.edu

**Keywords:** *OPRM1*, GPCR, alternative splicing, mu opioid receptor, opioid, morphine, fentanyl, biased signaling, gene targeting, pain

## Abstract

Most opioid analgesics used clinically, including morphine and fentanyl, as well as the recreational drug heroin, act primarily through the mu opioid receptor, a class A Rhodopsin-like G protein-coupled receptor (GPCR). The single-copy mu opioid receptor gene, *OPRM1*, undergoes extensive alternative splicing, creating multiple splice variants or isoforms via a variety of alternative splicing events. These *OPRM1* splice variants can be categorized into three major types based on the receptor structure: (1) full-length 7 transmembrane (TM) C-terminal variants; (2) truncated 6TM variants; and (3) single TM variants. Increasing evidence suggests that these *OPRM1* splice variants are pharmacologically important in mediating the distinct actions of various mu opioids. More importantly, the *OPRM1* variants can be targeted for development of novel opioid analgesics that are potent against multiple types of pain, but devoid of many side-effects associated with traditional opiates. In this review, we provide an overview of *OPRM1* alternative splicing and its functional relevance in opioid pharmacology.

## 1. Introduction

The actions of most clinically used opioid analgesics, such as morphine and fentanyl, are complex. Although they provide potent analgesia, they produce many undesirable side-effects such as respiratory depression, constipation, pruritus, physical dependence, and addiction. Misusing these opioids often leads to development of opioid use disorder, contributing significantly to the worldwide opioid epidemic. These opioid drugs act primarily on the mu opioid receptor (MOR) that was initially proposed by Martin in 1967 [1], established by pharmacological studies in 1973 [2,3,4], and verified by molecular cloning in 1993 [5,6,7]. Clinically, patients often showed divergent responses to mu opioids not only in analgesia but also in side-effects, and incomplete cross-tolerance among mu opioids led to the clinical practice of opioid rotation. These observations suggest the presence of multiple mu opioid receptors, a concept that was originally proposed by early pharmacological studies [8,9,10] and evolved by recent molecular studies of the mu opioid receptor gene, *OPRM1* [11,12].

Only a single-copy *OPRM1* gene has been identified, raising the hypothesis that alternative pre-mRNA splicing of the *OPRM1* gene generates multiple mu opioid receptors, contributing to the diversity and complexity of opioid responses seen in humans and animals. Driven by this hypothesis, many efforts have been made to isolate alternatively spliced *OPRM1* variants. We now know that the single-copy *OPRM1* gene undergoes extensive alternative splicing, creating an array of splice variants that are far more complicated than those defined by the early pharmacological studies. Conservation of *OPRM1* alternative splicing from rodents to humans suggests evolutionary and functional importance of the *OPRM1* gene. These *OPRM1* splice variants can be categorized mainly into three structurally distinct types: (1) full-length 7 transmembrane domain (TM) C-terminal variants; (2) truncated 6TM variants; and (3) truncated single TM variants, all of which are conserved from rodents to humans. Accumulated evidence indicates that these *OPRM1* splice variants play important roles in mu opioid pharmacology. This review outlines the progress regarding *OPRM1* alternative splicing and its pharmacological functions.

## 2. Mechanisms and Functions of Alternative Pre-mRNA Splicing

RNA splicing or precursor mRNA (pre-mRNA) splicing discovered in 1977 [13,14] involves removing introns and connecting exons to produce mature mRNA, a post-transcriptional event in metazoan organisms essential for many biological and pathological processes. RNA splicing is mediated through the spliceosome, a large complex including five small nuclear ribonucleoprotein particles (snRNPs, U1, U2, U4/U6, and U5) and more than 150 proteins. Alternative splicing is a major source of generating proteomic diversity. It has been estimated that ~95% of human multiple-exon genes undergo alternative splicing [15,16,17]. There are several types of alternative splicing, including exon skipping, alterative 5′ or 3′ splicing, intron retention, mutually exclusive exons, and alternative promoters and poly(A) sites [15]. Alternative splicing is differentially regulated in various tissues and cell types, as well as in different developmental stages, providing its diverse functions in a spatial and temporal manner. The magnitude of these effects varies from very subtle modifications to a complete loss or gain of function. Alternative splicing also participates in RNA processing itself, from pre- to post-transcriptional events [18].

Extensive studies of alternative splicing in a number of genes, including the c-src [19,20,21], Troponin T [22,23,24], and FGFR2 [25,26,27], elucidated mechanisms of alternative splicing. Our understanding of alternative splicing mechanisms was further extended by the identification of RNA binding proteins that modulate alternative splicing, such as the serine/arginine-rich (SR) family proteins [28,29,30,31], hnRNPs [32,33,34], the CELF protein family [35,36,37], and the neuro-oncological ventral antigen (NOVA) family [38,39,40]. In general, alternative splicing is a process of coordinating specific cis-acting elements in pre-mRNA and certain trans-acting factors with the basal spliceosome.

Aberrant alternative splicing has been associated with a number of human diseases [41,42], such as amyotrophic lateral sclerosis [43,44], paraneoplastic neurological disorders [38,45], myotonic dystrophy [46], spinal muscular atrophy [47,48], and cancer [27,49,50,51,52]. Antisense technologies have been used to correct aberrant alternative splicing for treating several human diseases. Spinraza (nusinersen), an antisense oligonucleotide targeting exon 7 of survival of moter neuron 2 (*SMN2*) gene, was the first FDA-approved drug to treat spinal muscular atrophy.

## 3. Alternative Splicing in GPCRs

There are about 800 different G protein-coupled receptors (GPCRs), representing the largest group of transmembrane receptors encoded by the human genome. GPCRs are targets for ~35% of FDA approved drugs [53,54]. Structurally, canonical GPCRs are characterized by an extracellular N-terminus, followed by seven transmembrane (TM) α-helices (TM-1 to TM-7) connected by three intracellular and three extracellular loops, and finally an intracellular C-terminus. GPCRs mediate diverse physiological responses, such as sense of vision, smell, taste, and nociception, by transducing extracellular stimuli to intracellular signaling molecules, such as heterotrimeric G proteins and β-arrestins.

Approximately 50% of GPCRs are subject to alternative pre-mRNA splicing [55], producing structurally diverse GPCR isoforms or variants. These splice variants can have alternative N- or C-termini, variable extracellular or intracellular loops, and truncated transmembrane domains, as well as alternative 5′ or 3′ untranslated regions (UTR). For example, genomic data search of non-olfactory receptor GPCR genes predicts ~12% of human GPCR genes have one or more alternatively spliced C-terminal variants [56], while over 9 and 7% of human GPCR genes contain truncated 6TM and single TM variants, respectively [57,58]. Alternative splicing of GPCR genes adds additional layers of complexity to the regulation and function of GPCR pharmacology. Table 1 lists selected human GPCRs with known splice variants along with their tissue distribution, and agonist/antagonist. However, the function of a majority of GPCR splice variants remains unknown.

## 4. Alternative Splicing of the Mu Opioid Receptor Gene, *OPRM1*

Most opioid analgesics used clinically, including morphine and fentanyl, as well as the recreational drug heroin, act primarily through mu opioid receptors. The mu opioid receptor was first proposed as the M (morphine, µ) receptor, together with the N (nalorphine, probably corresponding to the kappa_1_ receptor) receptor, by Martin in 1967, based on the pharmacological studies of nalorphine [1]. Establishment of opioid receptor binding assays [2,3,4] and identification of endogenous opioid peptides [86,87,88,89,90] facilitated the pharmacological defining of opioid receptor subtypes as mu, delta, and kappa receptors. Molecular cloning of the delta opioid receptor (DOR-1) [91,92] quickly led to isolation of the mu opioid receptor (MOR-1) [5,6,7] and kappa_1_ opioid (KOR-1) [93,94], as well as KOR-3 or opioid receptor-like receptor (ORL-1) [95,96,97,98,99]. All these opioid receptors belong to the GPCR Class A rhodopsin family.

### 4.1. Concept of Multiple Mu Opioid Receptors

The concept of multiple mu opioid receptors was originally suggested based on clinical observations that different patients displayed divergent responses to mu opioids not only in analgesia, but also in side-effects and that incomplete cross-tolerance among mu opioids led to the clinical practice of opioid rotation. Similar observations were seen in animal models. Different inbred mouse strains showed various sensitivities to mu opioids. For example, the 129/P3 inbred mouse strain is resistant to produce morphine tolerance and naloxone-precipitated withdrawal, sharply contrasting with the robust morphine tolerance and physical dependence seen in C56BL/6J and SWR/J inbred mouse strains [100,101]. Multiple mu binding sites were first demonstrated with opioid receptor binding assays using [^3^H]-labeled opioids in 1981 [9], subsequently validated by computer modeling approaches [102,103] and visualized by autoradiography [104,105,106]. Further pharmacological studies using two antagonists, naloxazone and naloxonazine, defined two mu subtypes, mu_1_ and mu_2_ [10]. A third mu subtype, morphine-6β-glucuronide (M6G) receptor, was proposed based on differences in analgesia potency, binding affinity of morphine, morphine analgesia in CXBK mice, and sensitivity toward this.

3-*O*-methylnaltrexone, a mu antagonist, had high affinity toward M6G and blocked analgesia of M6G, but not morphine [10,107,108,109]. Subsequent antisense oligonucleotide and *Oprm1* knockout studies provided molecular insights into the M6G site (see below) [107,110,111,112]. The identification of the MOR-1 sequence and its gene, *OPRM1*, allows us to explore multiple mu opioid receptors at the molecular level. The concept of multiple mu opioid receptors has been evolved by isolation and characterization of an array of alternatively spliced variants or isoforms from the single-copy *OPRM1* gene. The correlation between mu subtypes such as mu_1_ and mu_2_ defined by the early pharmacological studies and the *OPRM1* splice variants remains unknown, although several hypotheses such as heterodimerization were speculated.

### 4.2. Evolution of OPRM1 Gene

All four opioid receptor genes, including mu receptor (*OPRM1*), delta receptor (*OPRD1*), kappa receptor (*OPRK1*), and opioid-receptor like (*OPRL1*), are found only in vertebrates. As suggested by phylogenetic studies, the opioid receptor genes were generated through a two-round genomic duplication (paleoploidization) from a single ancestral opioid receptor gene (unireceptor) [113,114,115]. The first-round duplication produced *OPRD1*/*OPRM1* and *OPRL1*/*OPRK1* genes, which were subsequently segregated into individual *OPRD1*, *OPRM1*, *OPRL1,* and *OPRK1* [116,117,118]. Like the evolutionary tree of life, analyzing MOR-1 protein sequences predicted from 27 species define four major clades as follows: (1) fish, (2) amphibians, (3) birds, and (4) mammals [119]. Sequence alignments of MOR-1 proteins from different species reveal that the high-homology regions are located in the transmembrane domains and the three intracellular loops, suggesting the importance of these conserved structures in mu ligand binding and G protein coupling.

The genomic structure of the *OPRM1* gene appears to have evolved along with the four polygenetic clades [12,120]. In fish (teleosts), the *OPRM1* has five exons and each exon encodes regions with related TM domains: exon 1 (TM1), exon 2 (TM2–TM4), exon 3 (TM5), exon 4 (TM6), and exon 5 (TM7). In amphibians, the last three exons were merged into a single third exon encoding TM5–TM7. This structure with three exons encoding 7TMs has been maintained throughout birds and mammals. Starting in the chicken, exon 4 that encods 12 amino acids at the intracellular C-terminal tail evolved, which is maintained in all the mammals. Additional 5′ and/or 3′ exons such as exons 7 and 11 were integrated in mannals as well, providing a genomic structure for complex alternative splicing of the *OPRM1* gene. The functional importance of exon 7- and exon 11-associated splice variants in opioid pharmacology is discussed below.

### 4.3. OPRM1 Gene Structure

The chromosonal location of the mouse and human *OPRM1* gene was initially mapped to chromosomes 10 (10 cM from the centromere) and 6 (6q24-25), respectively, using traditional chromosomal mapping approaches [121,122,123,124], soon after MOR-1 cDNA was identified. Genomic sequencing initiated by the Human Genome Project confirmed the early *OPRM1* mapping results, and revealed the precise chromosomal locations of the *OPRM1* gene not only in mice and humans, but also in over 30 species. All the species contain a single copy of *OPRM1* gene. Genomic sequencing also showed that genes adjacent to *OPRM1* are conserved [11,120]. For example, the regulator of G protein signaling 17 (RGS17) gene is located at the upstream of the *OPRM1* gene, while the interaction protein for cytohesin exchanges factors 1 (IPCEF1) gene and subunit 5 of the splicing factor 3b (SF3b5) gene are found downstream of the *OPRM1* gene. Interestingly, a 70 bp sequence of exon 7 or exon *O* in the *OPRM1* gene overlaps with the IPCEF1 gene in the reversed orientation, raising questions about their interaction and regulation at transcription and posttranscription levels [11].

The *OPRM1* gene was initially identified as a structure with four exons and three introns using the original MOR-1 cDNA probe [125]. Mapping of additional exons at the 5′ and 3′ of the *OPRM1* genes identified in alternatively spliced transcripts further extends the *OPRM1* genes. Currently, the mouse, rat and human *OPRM1* genes have approximate 270, 264, and 210 kb, respectively [11,12].

The *OPRM1* gene has two independent promoters, exon 1 (E1) and exon 11 (E11) promoters, which are conserved from rodents to humans. E1 promoter was identified as a dual-promoter model that contains a proximal promoter and distal promoter [126,127,128]. E1 promoter lacks a TATA box but has several cis-acting elements such as GC-rich elements with multiple transcription start points, suggesting a housekeeping gene mode. E11 promoter, however, has a TATA box using a single transcription start point, favoring an eukaryote class II promoter mode. Each promoter controls expression of a unique set of splice variants. Differential expressions of the two promoters in brain regions such as hippocampus and substantia nigra in a transgenic mouse model suggest region-specific promoter activity [129].

### 4.4. Alternatively Spliced OPRM1 Variants

The *OPRM1* splice variants can be categorized into three major types based on receptor structure: (1) full-length 7TM C-terminal variants; (2) truncated 6TM Variants; and (3) single TM variants [11,12] (Figure 1, Figure 2 and Figure 3).

#### 4.4.1. Full-Length 7TM C-Terminal Splice Variants

Full-length C-terminal variants are generated through *OPRM1* 3′ splicing, which mainly refers to splicing from exon 3 to its downstream exons. All of the full-length 7TM C-terminal variants share the same exons 1/2/3 that encode the majority of the receptor, including the N-terminus, 7TM domains, three intracellular and three extracellular loops, and part of intracellular C-terminus, except for a distinct C-terminal tail encoded by an alternative exon(s) produced from the 3′ splicing.

The first two full-length 7TM C-terminal variants, hMOR-1A [130] and rMOR-1B [131], were identified in human and rat, respectively. Subsequent 3′RACE using anchor primers from exon 3 isolated many alternative 3′ exons and further PCR cloning generated full-length C-terminal variant sequences. To date, 22 full-length 7TM C-terminal variants in mice, 13 in rats, and 12 in humans have been isolated (Figure 1, Figure 2 and Figure 3) [132,133,134,135,136,137,138,139,140]. The predicted amino sequences from each of the 7TM C-terminal variants are unique and the lengths of the C-terminal tail sequences vary from the shortest with one amino acid in hMOR-1B4 [136] and rMOR-1D [137] to the longest with 88 amino acids in mMOR-1U [138] (Table 2). Moreover, several potential phosphorylation sites, such as protein kinase C, GRKs, cAMP- and cGMP-dependent protein kinase, and Casein kinase II, were identified in these C-terminal tails. The functional importance of these 7TM C-terminal variants in opioid pharmacology has been demonstrated using in vitro cell models showing differences in mu agonist-induced G protein coupling, phosphorylation, internalization and post-endocytic sorting and by differential region- and cell-specific expression, as well as suggested by findings using C-terminal truncation mouse models (see below).

Several C-terminal tail sequences are highly conserved from rodents to humans. For example, exon 3b in MOR-1A, an intron retention variant, encodes identical four amino acids (VCAF) in mouse MOR-1A (mMOR-1A) [142] and human MOR-1A (hMOR-1A) [130]. Exon 5a encodes the same five amino acids (KIDLF) in mMOR-1B1 and rMOR-1B1, which matches exactly to the first five amino acids encoded by human exon 5a in hMOR-1B1. However, hMOR-1B1 has an additional nine amino acids beyond the conserved five amino acids. Human exon *O* in hMOR-1*O* is a homolog of mouse exons 7a7b in mMOR-1*O*. Both exons predict an intracellular C-terminal tail with 30 amino acids share 63% of identity [137].

#### 4.4.2. Truncated 6TM Variants

Searching alternative 5′ exons of the mouse *Oprm1* gene using a modified 5′RACE approach led to identification of four new exons including exons 11, 12, 13, and 14 [143]. Both exons 11 and 12 were mapped to ~30 kb and 28 kb upstream of exon 1, respectively, while exons 13 and 14 were located between exons 1 and 2. Alternative splicing from exon 11 to its downstream exons generates nine exon 11-associated variants, five of which, mMOR-1G, mMOR-1K, mMOR-1L, mMOR-1M, and mMOR-1N, are truncated 6TM variants, in addition to three full-length 7TM variants, mMOR-1H, mMOR-1i, and mMOR-1J, and one single TM variant mMOR-1T (Figure 2). Splicing from exon 11 to exon 2 by skipping exon 1 generates three 6TM variants mMOR-1G, mMOR-1M, and mMOR-1N, which lack the first TM encoded by exon 1 [143]. Since the mouse exon 11 encodes 27 amino acids that do not predict a transmembrane domain, translation from exon 11 AUG in frame with exon 2 yields a receptor with 6TM (TM2–TM7) from exons 2 and 3. Splicing from exon 11 to exons 13 and 14 also by skipping exon 1 generates two 6TM variants, mMOR-1K and mMOR-1L. Although translation from exon 11 AUG predicts a short peptide beyond the 27 amino acids from exon 11, both mMOR-1K and mMOR-1L can produce a 6TM receptor by using an alternative AUG in exon 2.

Identification of the mouse exon 11-associated variants quickly led to isolation of the mouse homologs of exon 11-associated variants in rats and humans (Figure 1 and Figure 3) [144,145]. Based on the genomic databases, rat and human exon 11 are found at ~21 and ~28 kb upstream of their exon 1, respectively, a genomic location similar to that of the mouse exon 11 (~30 kb). PCR cloning using primers anchoring the respective exon 11 homologs and downstream exons identified four exon 11-associated variants in humans and seven in rats [144,145]. Like mMOR-1G, splicing from exon 11 to exon 2 generates two 6TM variants, rMOR-1G1, and rMOR-1G2 or hMOR-1G1 and hMOR-1G2, in both rats and humans. However, both rat and human exon 11 have an alternative 5′ splice site, which splits exon 11 as exons 11a and 11b. Splicing from exon 11a to exon 2 produces rMOR-1G2 or hMOR-1G2. The rat and human exon 11a predict 7 and 16 amino acids, respectively, which are in frame with exon 2. Thus, both rMOR-1G2 and hMOR-1G2 encode a 6TM receptor translated from the exon 11a AUG without the first TM. On the other hand, splicing from exon 11b to exon 2 yields rMOR-1G1 and hMOR-1G1. Similar to mMOR-1K and mMOR-1L, both rMOR-1G1 and hMOR-1G1 predict a 6TM receptor when the AUG in exon 2 is used, and produce a short peptide sequence if exon 11a AUG is used. Splicing from exon 11 to exon 1 produces an additional five variants in rats and two variants in humans, all of which encode a 7TM receptor when translated from exon 1 AUG. The functional relevance of the 6TM variants has been implicated in mediating the actions of a new type of opioid analgesics that are potent against a spectrum of pain models without many side-effects associated with traditional opiates [12] (see below).

An additional human 6TM variant hMOR-1K was isolated based on the finding of a mouse exon 13 homolog in the human *OPRM1* gene [146]. Unlike mMOR-1K that is associated with exon 11, hMOR-1K did not have exon 11 and its expression appeared to be controlled by its own promoter located at upstream of the human exon 13. Another human 6TM variant was identified as mu3 receptor with an exon composition of 2/3/new exon [147]. Both hMOR-1K and mu3 variants predict a 6TM receptor when translated using the exon 2 AUG, a scenario similar to rMOR-1G1, hMOR-1G1, mMOR-1K, and mMOR-1L (Figure 1).

#### 4.4.3. Single TM Variants

All of the single TM variants contain exon 1 that encodes the entire N-terminus and the first TM. Alternative splicing from exon 1 to downstream exons by exon skipping or insertion causes a reading-frame shift to terminate translation early, predicting a truncated receptor with a single TM. Currently, there are five single TM variants in mice, two in rats, and four in humans (Figure 1, Figure 2 and Figure 3). The first two single TM variants, hMOR-1S and hMOR-1Z, were isolated from the human *OPRM1* gene [11,85,148]. Splicing from exon 1 to exon 4 by skipping exons 2 and 3 produces hMOR-1S, while hMOR-1Z has exons 1/3/4 by skipping exon 2. Both MOR-1S and MOR-1Z were subsequently identified in mice and rats with the same splicing mechanisms [85]. An additional two human single TM variants were isolated as SV1 and SV2 in which an exon A or B was inserted between exons 1 and 2, respectively [149]. Early translation termination in exon A or B results in a truncated single TM receptor.

The mouse *Oprm1* gene has another three single TM variants, mMOR-1Q, mMOR-1R, and mMOR-1T [85]. Both mMOR-1Q and mMOR-1R have exon 2 skipping, but contain a different exon composition downstream of exon 3. In mMOR-1T, there is an exon 16 insertion between exons 1 and 2, predicting a single TM variant. Although the single TM variants cannot bind any opioids, their roles were shown in modulating the expression and function of the full-length 7-TM mu opioid receptor (MOR-1) (see below) [85].

## 5. Expression and Function of the *OPRM1* Variants

### 5.1. Expression of the *OPRM1* Variant mRNAs and Proteins

#### 5.1.1. Region-Specific and Strain-Specific Expression of the *OPRM1* Variant mRNAs

Northern blot analysis was initially employed to determine mRNA expression of *OPRM1* variants in the brain using individual exon probes or combined exon probes. Multiple bands with different sizes and intensities were observed [6,123,132,134,135,136,137,143,150], suggesting *OPRM1* alternative splicing. However, it is difficult to examine mRNA expression of individual splice variants using exon probes in Northern blot analyses because some exons are shared among several splice variants. Semi-quantitative RT-PCR assays were designed to investigate the expressions of the *OPRM1* variant mRNAs in a number of dissected brain regions, including the cortex, thalamus, striatum, hypothalamus, hippocampus, brain stem, cerebellum, and spinal cord [132,133,143,144]. For example, high level of mMOR-1C mRNA but low levels of mMOR1D and mMOR-1E mRNAs were expressed in the mouse thalamus [132]. In human brain, hMOR-1G2 was highly expressed in the prefrontal cortex, piriform cortex, nucleus accumbens, and pons, but not in the temporal cortex and spinal cord [144].

Using real-time quantitative PCR (qPCR), a more accurate method measuring mRNA levels, the expression profiles of thirty *Oprm1* splice variant mRNAs in ten different brain regions of four inbred mouse strains, C56BL/6J, 129P3/J, SJL/J, and SWR/J [151], and CD-1 mice [152] were examined. The results showed marked differences of the variant mRNA expression among the brain regions or among the mouse strains, suggesting region-specific or strain-specific alternative splicing of the *OPRM1* gene. For example, 7TM C-terminal variant mMOR-1A was expressed highly in the hypothalamus and thalamus but lowly in the striatum and prefrontal cortex in SJL/J mice. In the same PAG region, mMOR-1G, a 6TM variant, was expressed much higher in C57Bl/6J mouse strain than in the other three mouse strains.

Among the three types of *Oprm1* variants, the overall expressions of all 7TM variants were dominant throughout the brain regions and mouse strains, while the overall expressions of all 6TM and 1TM variants were lower than those of all 7TM variants (Table 3) [153]. However, all 6TM and 1TM variants were expressed at relatively high levels in certain regions or mouse strains. For example, in the PAG region of C57BL/6J mice, the expression level of all 6TM variants was 20.4% of all 7TM variants, which was the highest among the brain regions and among the mouse strains. The expression level of all 1TM variants in the prefrontal cortex of SJL/J mice was higher than those in the other three mouse strains. These results further demonstrated region-specific and strain-specific alternative splicing of the *Oprm1* gene.

Altered expression of the *OPRM1* splice variants were seen in multiple brain regions in a morphine tolerance mouse model [152], the human brain of HIV-infected individuals [153,154], and the medial prefrontal cortex of human heroin abusers and heroin self-administering rats [155]. Sex difference in expression of the *Oprm1* variant mRNAs among the mouse brain regions was also observed [156].

#### 5.1.2. Region-Specific and Cell-Specific Expression of the *OPRM1* Variant Proteins

The distribution of *OPRM1* variant proteins was investigated using several antisera generated against individual exon coding sequences [157,158,159,160]. These studies revealed dramatic differences in the regional distribution of the exon-containing *OPRM1* variant proteins. For example, the distribution of exons 7/8-like immunoreactivity (LI) detected with an antisera against a peptide from mouse exons 7/8 clearly differed from that of exon 4-LI using an antisera against a peptide from exon 4 in several regions, such as the medial eminence, thalamic nuclei, and nucleus ambiguous [158]. The two antisera also labeled different cell populations, as demonstrated in the laminae layer I–II of the spinal cord [158]. An antisera raised against an exon 8 peptide labeled uniquely in the dentate gurus, the mossy fibers of the hippocampal formation and the nucleus of the solitary tract [157], while an antisera raised against exon 3/5a hybridized predominantly in the olfactory bulb [161]. The localization of variant proteins was also examined at the ultrastructural level [162]. For example, in the superficial laminae of the spinal cord, the exon 7/8-LI was predominantly distributed in presynaptic cells and co-localized with calcitonin gene-related peptide (CGPR), whose antibody labels unmyelinated primary afferents, in contrast to the exon 4-LI, which displayed an equal distribution between presynaptic and postsynaptic membranes and did not co-localize with CGPR [162].

It should be noted that several variant proteins encoded by the same exon(s) commonly share identical epitope(s), thus it is challenging to distinguish each individual variant at the protein level using exon-specific antisera by immunohistochemistry. For example, antisera against a peptide from mouse exon 4 would detect at least five exon 4-containing variants in mice including mMOR-1, mMOR-1H, mMOR-1I, mMOR-1J, and mMOR-1G. Similarly, exon 8-LI would correspond to mMOR-1D and mMOR-1N, whereas exon 7/8-LI would represent two variants, mMOR-1C and mMOR-1M. These studies also revealed the discrepancies between mRNA and protein levels for some of the variants. For example, in the mouse brain, the protein level of mMOR-1 detected by the exon 4-specific antisera was found to be comparable to that of mMOR-1C and mMOR-1M as determined by the exon 7/8-specific antisera [158], while the expression level of mMOR-1 mRNA was much higher than that of both mMOR-1C and mMOR-1M mRNA [132,143]. Although different antisera may differ in sensitivity, these studies reiterated the poor correlation between mRNA and protein levels in the brain and highlighted the importance of examining the expression of variants at both the mRNA and proteins expression levels.

### 5.2. The Function of Full-Length 7TM C-Terminal Variants

#### 5.2.1. Opioid Receptor Binding Affinity

To determine the binding profiles of full-length 7TM C-terminal variants, opioid receptor binding assays using [^3^H] DAMGO, a full mu agonist, were performed with membranes from CHO cell lines that stably express each individual full-length C-terminal variant [132,133,134,135,136,137,142]. The results from saturation studies indicated that all the 7TM C-terminal variants had high affinity toward [^3^H] DAMGO, which was anticipated since they all share the same binding pocket. Competition studies showed mu selectivity with a number of mu opioids such as morphine, M6G, and naloxone for all the 7TM C-terminal variants. However, several endogenous opioid peptides revealed different binding affinities among the 7TM C-terminal variants. For example, both β-endorphin and dynorphin A were over four times more potent in mMOR-1D than in mMOR-1 [132]. These results raise intriguing questions of how different C-terminal tails impact on the binding pocket, specifically toward the endogenous opioid peptides.

#### 5.2.2. Mu Agonist-Induced G Protein Coupling, Internalization, Phosphorylation, and Post-Endocytic Sorting

The intracellular C-termini of GPCRs play an important role in the signaling transduction system, especially the G-protein activation. Mu agonist-induced G protein activation in the *OPRM1* 7TM C-terminal variants were extensively studied using membranes from CHO cells stably expressing the individual variant in [^35^S] GTPγS binding assay, a commonly used method to assess the overall agonist-induced receptor-G-protein coupling. The results revealed marked differences in mu agonist-induced G-protein activation in both potency, measured by the EC_50_ values, and efficacy, determined by the maximal stimulation, among the 7TM C-terminal variants in mice, rats, and humans [135,136,137,163]. Several mu agonists displayed varied efficacies among various 7TM C-terminal variants. For example, morphine and M6G were more efficacious in hMOR-1A than hMOR-1B1, hMOR-1B3, and hMOR-1B5, while β-endorphin was a full agonist for hMOR-1B5 and a partial agonist for hMOR-1A [136]. There was lack of correlation between the binding affinity (*K*_i_) and potency (EC_50_). β-endorphin bound to both rMOR-1 and rMOR-1C1 with similar affinity. However, β-endorphin was over 20 times more potent in rMOR-1 than in rMOR-1C1 in [^35^S] GTPγS binding. Also, there was little correlation between potency (EC_50_) and efficacy. Together, these results demonstrated that different intracellular C-terminal tails in the 7TM full-length variants have significant influence on mu agonist-induced G protein activation.

Mu agonist-induced MOR-1 phosphorylation involves receptor desensitization and internalization, contributing to the development of mu opioid tolerance and physical dependence [164,165,166,167]. Several mu agonist-induced phosphorylation sites in the intracellular C-terminus encoded by exon 3 in MOR-1, such as ^354^TSST^357^ and ^375^STANT^379^ motifs, were identified [168,169,170]. Involvement of these phosphorylation sites in mu opioid actions such as analgesia, tolerance, respiratory depression, and constipation was demonstrated using several phosphorylation-deficient mouse models [171]. However, these phosphorylation sites are shared by all the 7TM C-terminal variants. The potential phosphorylation sites for several kinases such as GRKs, protein kinase C, casein kinase, and tyrosine kinase predicted from alternatively spliced C-terminal tails raise questions regarding their responses to mu agonists (Table 2). Koch et al. showed that morphine-induced receptor phosphorylation in mMOR-1D and mMOR-1E was robust, sharply contrasting with the nonresponsiveness in mMOR-1 and mMOR-1C [172], suggesting additional morphine-induced phosphorylation sites in the C-terminal tails encoded by the respective C-terminal tails or modulation of overall receptor phosphorylation by the C-terminal tails. The differential phosphorylation profiles among these variants were correlated with morphine-induced internalization and resensitization [172]. Additionally, threonine 394 in the exon 4-encoded C-terminal tail of rMOR-1 was phosphorylated in response to DAMGO [173]. Interestingly, a mutant mouse with the single point mutation T394A showed loss of etorphine-induced MOR-1 internalization in the spinal dorsal horn neurons and diminished tolerance by morphine and etorphine, as well as increased vulnerability to heroin self-administration [174]. Furthermore, morphine induced robust internalization of exon 7-associated mMOR-1C, but not exon 4-associated mMOR-1, in the mouse lateral septum [175].

After being internalized by mu agonists, mMOR-1 is either migrated to the lysosome for degradation or recycled back to the plasma membrane. Exon 4-encoded C-terminal sequences contain a motif termed the MOR1-derived recycling sequence (MRS) that allows efficient post-endocytic sorting of mMOR-1 to the plasma membrane [176]. Absence of the exon 4-encoded MRS in mMOR-1B, mMOR-1D, and mMOR-1E led to rapid lysosomal degradation due to inefficient receptor recycling following DAMGO-induced internalization, indicating the importance of the C-terminal sequences in determining the post-endocytic sorting direction of GPCRs [177].

#### 5.2.3. Mu Agonist-Induced Biased Signaling at 7TM C-Terminal Variants

Biased signaling of GPCRs is defined based on observations that different agonists can produce divergent signaling transduction pathways through a single receptor. Biased signaling through G protein and β-arrestin 2 has been extensively studied in the original MOR-1 [168,178,179,180,181,182]. Identification of multiple *OPRM1* 7TM C-terminal variants opened questions about their roles in mu agonist-induced biased signaling. The effect of different 7TM full-length C-terminal variants on mu agonist-induced signaling bias between G protein coupling and β-arrestin 2 recruitment was studied using [^35^S] GTPγS binding and PathHunter β-arrestin 2 recruitment assay in the same CHO cells stably expressing individual 7TM C-terminal variants [183]. The results showed marked differences in mu agonist-induced G protein activation and β-arrestin 2 recruitment among the 7TM variants. For example, exon 7-associated 7TM variants, particularly mMOR-1*O*, displayed greater β-arrestin 2 bias for most mu agonists than exon 4-associated 7TM variant mMOR-1, suggesting that exon 7-associated variants favor mu agonist-induced β-arrestin 2 binding. This hypothesis was supported by the identification of a consensus phosphorylation code, PxPxxE/D or PxxPxxE/D, in exon 7-encoded sequences for high affinity arrestin binding proposed by homology modeling with the crystal structures of several GPCRs (Table 2) [141]. When this predicted phosphorylation code was mutated in mMOR-1*O*, DAMGO-induced β-arrestin 2 recruitment was greatly attenuated (our unpublished data) [183]. These results also suggested a mechanism of the in vivo function of exon 7-associated variants in several morphine actions observed in an exon 7 C-terminal truncation mouse model (mE7M-B6), which showed similar phenotypes in a β-arrestin 2 knockout (KO) mouse (see below) [83]. The finding that a single mu agonist can produce differential signaling through multiple 7TM C-terminal variants from in vitro cell line studies offers new insights into our understanding of biased signaling in vivo where all the 7TM C-terminal variants are co-expressed.

#### 5.2.4. In Vivo Function of 7TM C-Terminal Variants

##### Morphine-Induced Itch (Pruritus)

Opioid-induced itch (pruritus) is one of the most prevalent acute side effects of the spinal or epidural use of opioids in patients [184,185,186], which has limited the clinical use of opioids. However, the molecular mechanisms underlying opioid-induced itch remained largely unknown. Liu et al. reported that an *Oprm1* 7TM C-terminal splice variant mMOR-1D was the major player in spinal morphine-induced scratching in mice [187]. Downregulating the spinal mMOR-1D with siRNAs reduced morphine-induced scratching without affecting morphine analgesia. They further demonstrated that morphine-induced scratching was mediated through heterodimerization of mMOR-1D and gastrin-releasing peptide receptor (GRPR) in the spinal cord. Disrupting mMOR-1D/GRPR heterodimerization attenuated morphine-induced scratching, but not morphine-induced analgesia [187]. The same group further found that similar to mMOR-1D, one human 7TM C-terminal variant hMOR-1Y can physically associate with GRPR to function as an itch receptor [188]. These studies further established the pharmacological relevance of *OPRM1* 7TM C-terminal variants and also provided a potential therapeutic strategy to specifically alleviate itch without attenuation of opioid analgesia.

##### Morphine Tolerance, Dependence, and Reward

To explore the in vivo functions of alternatively spliced C-terminal tails, three C-terminal truncation mouse models were established in two inbred mouse strains, C57BL/6J (B6) and 129/SvEv (129), using a stop codon targeting strategy that truncates a designated exon at the translational level with limited effects on overall *Oprm1* transcription and alternative splicing (Figure 4) [83]. In the first model (mE3M), the C-terminal tails of all the variants are truncated at the end of E3 by a stop codon introduced at the end of E3. In the second (mE4M) and third models (mE7M), the E4- or E7-encoded C-terminal tails are truncated by inserting a stop codon at the beginning of E4 or E7, respectively [83]. Studies using these mice show divergent roles of the C-termini in various morphine-induced behaviors. For example, the E7-encoded C-terminal truncation in mE7M-B6 mice decreases morphine tolerance and reward without altering physical dependence, whereas the E4-encoded C-terminal truncation in mE4M-B6 mice accelerates morphine tolerance and reduces morphine dependence without affecting morphine reward (Figure 4) [83]. Moreover, in mE7M-B6 mutant mice, morphine-induced receptor desensitization in the hypothalamus and brain stem was lost, suggesting an E7-mediated mechanism underlying morphine tolerance [83].

The similarities found in several morphine-related behaviors and receptor desensitization between mE7M-B6 mutant and β-arrestin 2 KO mice suggest a physical and functional association of E7-encoded C-terminal tails with β-arrestin 2 [83,189,190,191,192]. This supposition was supported by our in vitro cell studies showing greater β-arrestin 2 bias for several mu opioids in the E7-associated variant mMOR-1*O* than in the E4-associated variant mMOR-1 [83,183]. However, several mu opioid–induced actions, including morphine analgesia, locomotion, and reward [193,194], and fentanyl and methadone tolerance (unpublished observation), observed in mE7M-B6 mice were unlike those detected in β-arrestin 2 KO mice, suggesting a β-arrestin 2-independent mechanism. Together, these results further highlight the functional relevance of the 7TM C-terminal variants, particularly exon 7-associated 7TM variants, in mu opioid pharmacology.

### 5.3. The Function of Truncated 6TM Variants

#### 5.3.1. Involvement of 6TM Variants in Heroin and M6G Analgesia

Although the truncated 6TM variants did not bind any available radiolabeled opioids when expressed in cell lines, their functional relevance was mainly revealed using gene targeting mouse models (Table 4). The pharmacological function of 6TM variants was first shown to involve the analgesic actions of heroin and M6G, but not morphine and methadone, by using two *Oprm1* KO mouse models. In an exon 1 knockout (E1 KO) mouse, morphine and methadone analgesia was completely lost [111]. However, heroin and M6G analgesia still was active despite a slightly reduced potency. Since knocking out E1 abolished all 7TM C-terminal variants and did not disrupt 6TM variants, these results suggested that 6TM variants involve heroin and M6G analgesia, while 7TM variants were responsible for morphine and methadone analgesia. This suggestion was further supported by the opposite phenotypes seen in E11 KO mice in which heroin and M6G analgesia were greatly decreased, while morphine and methadone analgesia were normal [112]. Furthermore, the analgesic actions of all mu opioids including morphine, M6G, and heroin were lost in a exon 2 deficient mouse in which both 7TM and 6TM variants were disrupted [111], consistent with the results from E1 KO and E11 KO.

#### 5.3.2. Essential Role of 6TM Variants in a Novel Type of Opioid Analgesic Drugs

The truncated 6TM variants were further discovered to mediate the actions of a novel opioid analgesic drug, 3′-iodobenzoylnaltrexamide (IBNtxA), which was derived from naltrexone [195,196]. Although the N-cyclopropyl group on the morphinan template leads to an antagonist pharmacophore, an example being naltrexone, incorporating an aryl amido group at the 6-position of the morphinan template leads to agonism at 6TM sites in the case of IBNtxA [195]. Similarly, such an incorporation in MP1104 confers its characterization as a kappa agonist at KOR-1 site [197], while MP1207 and MP1208 with the incorporation become mixed mu/kappa agonists [182]. Therefore, the 6-position arylamido group on the morphinan template puts the receptor in an agonist conformation. IBNtxA analgesia was lost in E11 KO mice, suggesting the role of exon 11-associated 6TM variants in IBNtxA analgesia. On the other hand, IBNtxA analgesia was fully active in a triple KO mouse in which only *Oprm1* 6TM variants remained, while all other opioid receptors including MOR 7TM variants, kappa opioid receptor (KOR-1), and delta opioid receptor (DOR-1) were knocked out, indicating that IBNtxA analgesia was independent of 7TM MOR variants, DOR-1, and KOR-1 [195]. Furthermore, intrathecal administration of lentivirus expressing individual 6TM variants, including mMOR-1G, mMOR-1M, mMOR-1N, mMOR-1K, and mMOR-1L, restored IBNtxA analgesia in a complete *Oprm1* E1/E11 KO mouse in which all 7TM, 6TM, and single TM variants were disrupted [84,198]. Together, these results demonstrate that 6TM variants are essential for IBNtxA analgesia. Additionally, the opioid receptor binding studies with [^125^I]BNtxA revealed a unique binding site in the triple KO mouse that was lost in both E11-KO and E2 KO mice [195,196,199,200]. Intriguingly, kappa-like drugs, such as ketocyclazocine and kappa_3_-like drug NalBzoH, competed for this site with high affinity, while many traditional mu, delta, and kappa opioids had very poor affinity for the site.

IBNtxA represents a novel class of opioid analgesics that are potent against multiple types of pain including thermal, inflammatory, and neuropathic pain, but do not produce many sides effects of traditional opioids, such as respiratory depression, physical dependence, and reward [195,201]. Therefore, the 6TM variants represent promising therapeutic targets for development of the novel analgesics with more favorable side-effect profiles.

#### 5.3.3. Function of 6TM Variants in Delta/Kappa Opioids and Non-Opioids

Using E11 KO and E1/E11 KO mice in combination with lentivirus rescue and antisense studies, Marrone et al. revealed the involvement of truncated 6TM variants in the analgesic actions of delta and kappa opioids, as well as α_2_-adrenergic drug [202]. The analgesia of DPDPE, a delta agonist, U50,488H, a kappa agonist and clonidine, a α_2_-adrenergic agonist, was lost in the E11 KO mice, consistent with the results from the antisense studies using an antisense oligo targeting E11 [202]. Re-expression of mMOR-1G, a 6TM variant, through lentivirus in E11 KO mice rescued the analgesia of DPDPE, U50,488H and clonidine [202], further confirming that the analgesia of delta, kappa and α_2_-adrenergic agonists is dependent on the 6TM variants. However, 6TM variants did not involve delta agonist-induced seizure, kappa agonist-induced aversion, and α_2_ agonist-induced hypolocomotion since these activities were still active in E11 KO mice [202]. Additionally, DPDPE analgesia was also dependent on *Oprm1* 7TM variants based on the observation that lentivirus-expressing mMOR-1G can only restore DPDPE analgeia in E11 KO mice, but not in the E1/E11 KO mice that lacked both 7TM and 6TM variants [202]. The 6TM variants demonstrated the ability to heterodimerize with β_2_-adrenergic receptors, providing a molecular mechanism of the analgesic synergy between the 6TM variants and β_2_-adrenergic drugs [203]. These studies expended the known functions of the 6TM variants beyond mu opioid receptors.

#### 5.3.4. A Chaperon-Like Function of 6TM Variants in Enhancing Expression of the 7TM MOR-1 at Protein Level

The function of the 6TM variants was also demonstrated in enhancing expression of the 7TM MOR-1 at the protein level through a chaperon-like function [204], a similar scenario seen in the single TM variants [85] (see below). When co-expressed with mMOR-1 in a Tet-Off inducible CHO cell line, mMOR-1G dose-dependently promoted the expression of the protein but not the mRNA of mMOR-1, as determined by opioid receptor binding and [^35^S]GTPγS binding. Using co-immunopreciptation and subcellular fractionation analysis, the study showed that the increased mMOR-1 receptor protein was mainly found in the plasma membrane, which was mediated through heterodimerization of mMOR-1G with mMOR-1 in the endoplasmic reticulum, suggesting a chaperone-like function [204].

#### 5.3.5. Role of 6TM Variants in Morphine Hyperalgesia

Several studies indicated that the 6TM variants were involved in morphine hyperalgesia. In the E11-KO mouse without any 6TM variants, morphine analgesia was fully retained, but morphine-induced hyperalgesia was diminished [112,205]. However, the triple KO mouse, which retained only the truncated 6TM variants and disrupted KOR1, DOR-1, and all the E1-associated 7TM MOR variants, lost morphine analgesia but retained morphine hyperalgesia [206]. Downregulating the 6TM variant mMOR-1K with intrathecal administration of a siRNA in CXB7/ByJ mice diminished morphine-induced hyperalgesia [207], suggesting involvement of mMOR-1K in morphine hyperalgesia. Together, these findings suggest that morphine analgesia is independent of truncated 6TM variants, whereas morphine hyperalgesia is dependent on truncated 6TM variants. In addition, truncation of 6TM variants in E11-KO mice led to reduction of morphine-induced hyperlocomotion and tolerance with no effect on morphine reward and respiratory depression [205].

### 5.4. The Function of Single TM Variants

It is intriguing that all the single TM variants contain the first TM encoded by exon 1, which could complement the 6TM variants that lack the exon 1. Although our unpublished data showed that the single TM variants can physically associate with the 6TM variants when co-transfected into CHO cells, the functional relevance of this physical association remains unknown. Similar to the 6TM variants, the functions of the single TM variants were indicated to increase expression of 7TM MOR-1 at the protein level through a chaperone-like mechanism [85]. Using a Tet-Off CHO system, Xu et al. showed that the single TM variants can dimerize with the 7TM MOR-1 in the endoplasmic reticulum to facilitate the proper folding of MOR-1 and reduce MOR-1 degradation through endoplamic reticulum-associated degradation (ERAD), leading to increased expression of MOR-1 protein. In vivo antisense studies further suggested the role of the single TM variants in morphine analgesia through modulation of 7TM MOR-1 expression [85].

Interestingly, the first TM domain of MOR-1 was also implicated in heterodimerization with DOR-1 to disrupt MOR-1/DOR-1 dimer using an engineered protein, MOR^TM1^-TAT (transactivator of transcription), resulting in enhanced morphine analgesia and reduced morphine tolerance [208]. This study raises questions on whether endogenously expressed single TM variants containing the first TM have a similar function.

## 6. Molecular Mechanisms Underlying *OPRM1* Alternative Splicing

Identification of multiple *OPRM1* splice variants raises questions regarding the molecular mechanisms underlying *OPRM1* alternative splicing. For example, what are cis-acting elements and trans-acting factors that control the *OPRM1* alternative splicing? Xu et al. initially identified an intronic single nucleotide polymorphism (SNP)(rs9479757) located near the 3′ of *OPRM1* exon 2 that associated with heroin addiction severity in a cohort of Chinese male heroin addicts [209]. Further characterizing this SNP using electrophoretic mobility shift assay (EMSA), minigene constructs, and RNA affinity purification coupled with proteomics led to the identification of a SNP-containing cis-acting element as an intronic splicing enhancer (ISE) that can interact with a splicing factor hnRNPH to regulate the exon 2 exclusion (skipping) or inclusion, and modulate expression of the full-length 7TM MOR-1 and 1TM variants hMOR-1S and hMOR-1Z [209]. The G allele of rs9479757 enhanced the binding of hnRNPH to this ISE, facilitating exon 2 inclusion to increase the expression of the 7TM hMOR-1 and reduce the expression of the 1TM hMOR-1S and hMOR-1Z, while the A allele had opposite effects. These results not only revealed the molecular mechanisms underlying *OPRM1* exon 2 exclusion/inclusion, but also demonstrated the function of rs9479757 in regulating *OPRM1* alternative splicing and contributing to heroin addiction severity. *OPRM1* alternative splicing is complex. Further deciphering the molecular mechanisms of *OPRM1* alternative splicing will advance our understanding of *OPRM1* gene regulation and function.

## 7. Conclusions

The early concept of multiple mu opioid receptors suggested by the pharmacological studies over 40 years ago has now been revolutionized by identifying an array of *OPRM1* alternatively spliced variants. These splice variants are generated through various splicing mechanisms such as alternative 3′ or 5′ splicing, intron retention, alternative promoters, or combinations of two or three mechanisms in a region-specific, cell-specific, or strain-specific fashion and conserved from rodents to humans. Based on the predicted transmembrane domains, these splice variants can be categorized mainly into three types: (1) full-length 7TM C-terminal variants, (2) truncated 6TM variants; and (3) truncated single TM variants. The functional relevance of these variants demonstrated by both in vitro cell lines and in vivo gene targeting animal models provides new insights into our understanding of complex mu opioid actions. More importantly, these *OPRM1* splice variants are potential therapeutic targets for developing novel analgesic drugs with more favorable side-effect profiles. Many GPCRs have predicted splice variants like those of the *OPRM1* gene. However, the functions of these variants remain largely unknown. Revealing the distinct functions of various *OPRM1* splice variants opens questions of whether alternative splicing may extend the pharmacological repertoire of GPCRs in general.

## Figures and Tables

**Figure 1 biomolecules-11-01525-f001:**
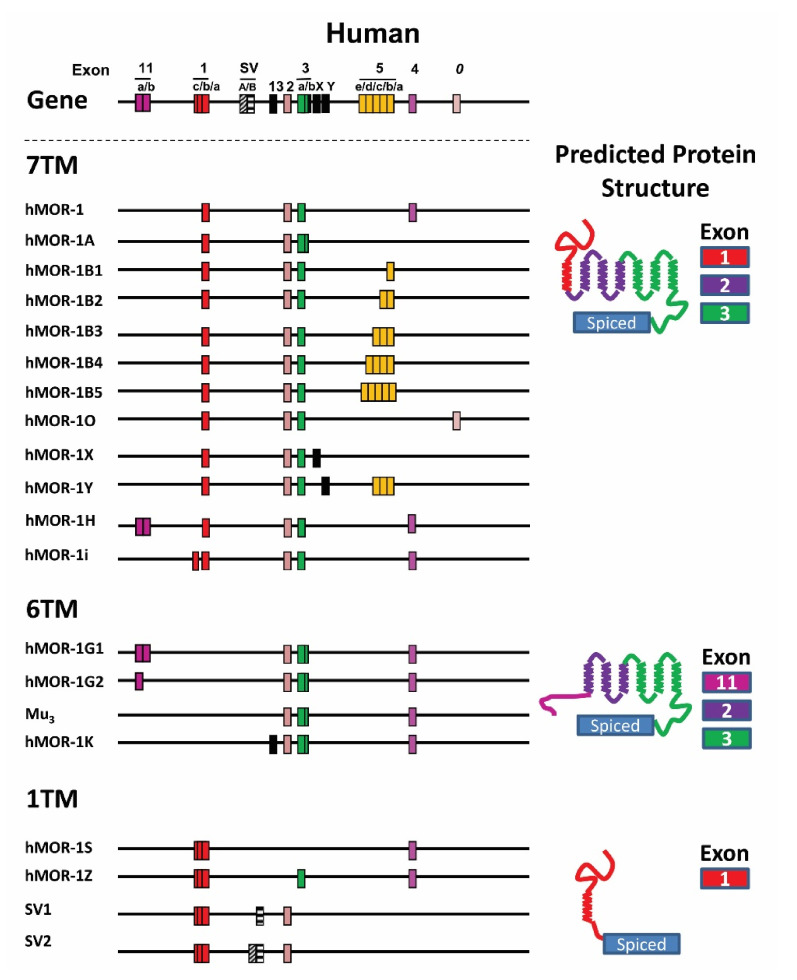
Schematic of the human *OPRM1* gene structure and alternative splicing. The top panel is the *OPRM1* gene structure. Alternatively spliced variants reported in the literature are grouped as 7TM, 6TM, and 1TM with predicted protein structure shown to the right and exon color coded to match the splicing schematic. From Pasternak and Pan 2013 [12].

**Figure 2 biomolecules-11-01525-f002:**
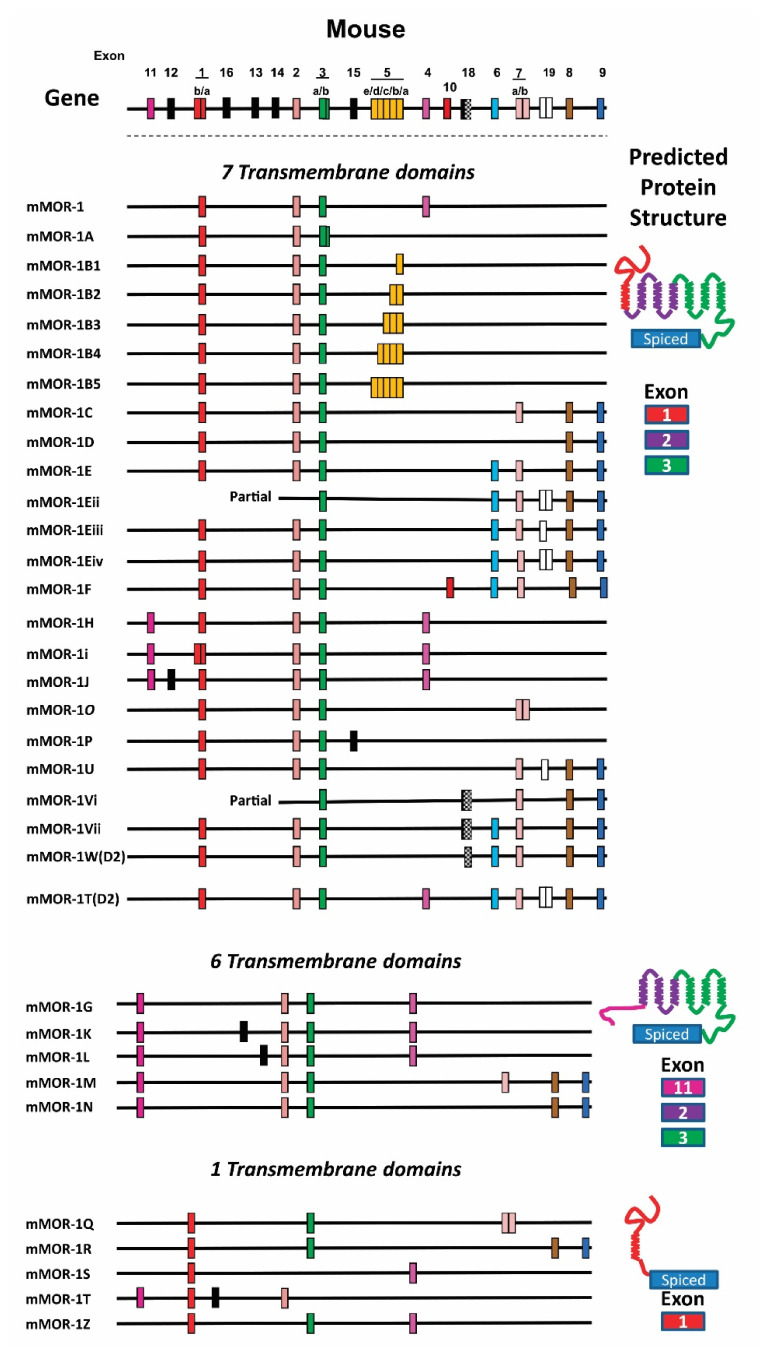
Schematic of the mouse *Oprm1* gene structure and alternative splicing. The top panel is the *Oprm1* gene structure. Alternatively spliced variants reported in the literature are grouped as 7TM, 6TM, and 1TM with predicted protein structure shown to the right and exon color coded to match the splicing schematic. From Pasternak and Pan 2013 [12].

**Figure 3 biomolecules-11-01525-f003:**
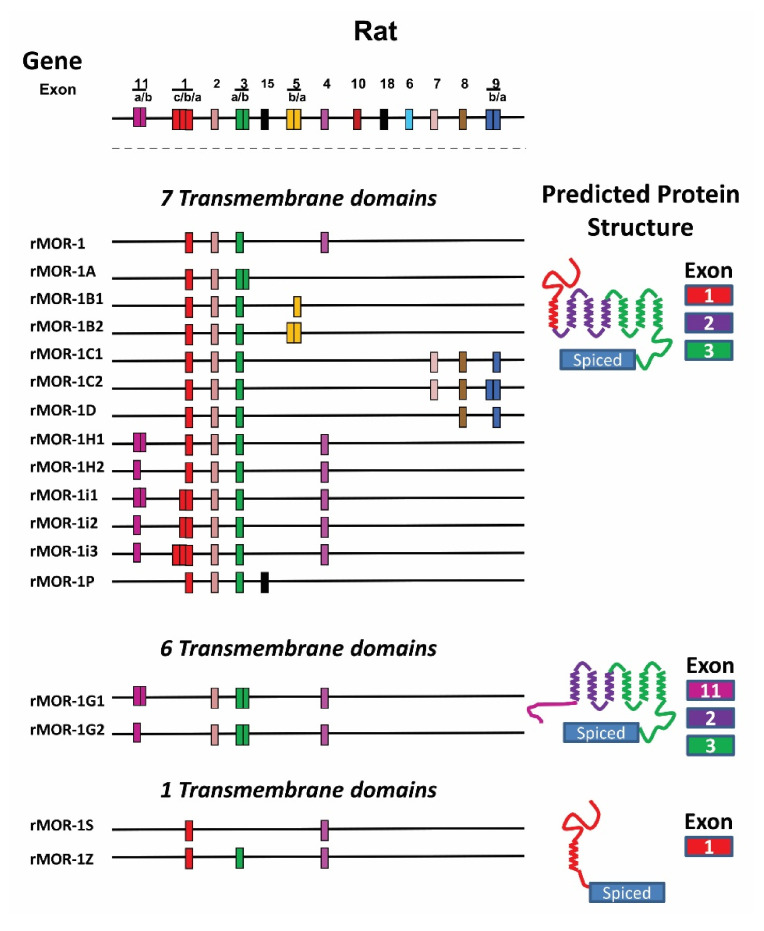
Schematic of the rat *Oprm1* gene structure and alternative splicing. The top panel is the *Oprm1* gene structure. Alternatively spliced variants reported in the literature are grouped as 7TM, 6TM, and 1TM with predicted protein structure shown to the right and exon color coded to match the splicing schematic. From Pasternak and Pan 2013 [12].

**Figure 4 biomolecules-11-01525-f004:**
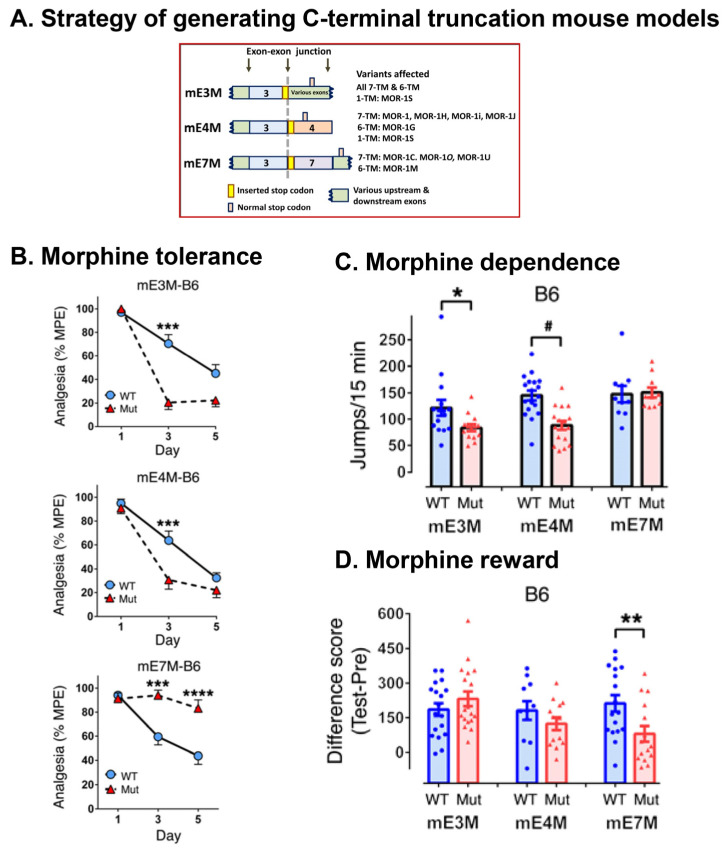
*Oprm1* C-terminal truncation mouse models adapted from Xu et al., 2017 [83]. (**A**). Schematic of the strategy generating C-terminal truncation mouse models. Three targeted mouse models, mE3M, mE4M, and mE7M, were generated by inserting a stop codon at an appropriate site within indicated exons shown by colored boxes. Inserted and original stop codons are indicated by yellow and pink bars, respectively. In mE3M, a stop codon was inserted at the end of exon 3 to eliminate every C-terminal tails of all 7TM and 6TM variants, as well as 1TM mMOR-1S. In mE4M and mE7M, a stop codon was created at the beginning of exon 4 or exon 7 to eliminate individual C-terminal tails encoded by exon 4 or exon 7 of indicated variants, respectively. (**B**). Morphine tolerance in the mutant mice on C57BL/6J (B6) background was induced by twice daily injections with morphine (10 mg/kg, s.c.) for 5 days. Morphine analgesia was determined 30 min after s.c. injection using a radiant-heat tail-flick assay. Results are shown as the percentage of maximum possible effect (% MPE). WT, wildtype mice; Mut, homozygous mice. ***: compared to WT, *p* < 0.001, 2-way ANOVA with Bonferroni’s *post hoc* test. (**C**). Morphine physical dependence was assessed on day 5 of chronic morphine treatment with naloxone (s.c.,1.0 mg/kg) injection 3 h after the last morphine treatment to precipitate withdrawal. The number of jumps within 15 min was used for the measurement of withdrawal. *: *p* < 0.05; #: *p* < 0.0001, compared with WT, 1-way ANOVA with Bonferroni’s *post hoc* test. (**D**). Morphine reward was assessed using a 6-day conditioned place preference protocol. **: *p* < 0.01, compared with WT, 1-way ANOVA with Bonferroni’s *post hoc* test. Figure (**A**–**D**) were adapted from Figure S2, Figure 1 and Figure 2 of Xu et al. 2017 [83], respectively.

**Table 1 biomolecules-11-01525-t001:** List of selected human GPCRs splice variants along with their agonist, antagonist, and tissue distribution.

GPCR Receptors	Gene Symbol	Receptor Name (Number of Variants)	Agonist	Antagonist	Tissue Distribution	Reference
Adrenergic Receptors	*ADRA1A*	α1A-AR C-term variant (4) 6-TM variant (11)	Dabuzalgron	Amitriptyline Promethazine Nortriptyline	Mainly liver, heart, and brain	[59,60,61,62,63]
	*ADRA1B*	α1B-AR: 6-TM variant (1)	Phenylephrine	Conopeptide p-TIA	Brain	[64]
Cannabinoid Receptors	*CNR1*	CB1 N-term variant (2)	THC 2-AG ACEA	TM38837 VD60	Mainly brain, DRG, and kidney	[65,66,67]
	*CNR2*	CB2 N-term variant (2)	THC 2-AG Nabilone	SR144528 AM-630	Mainly testis and spleen	[68]
Corticotropin-Releasing Hormone Receptor	*CRHR1*	CRH-R1 IL variant (1) C-term variant (1) N-term variant (2)	Stressin I Ovine CRH	Antalarmin	Mainly brain	[69]
	*CRHR2*	CRH-R2 N-term variant (3)	Urocortin II Urocortin III	K41,498	Mainly brain	[69]
Tachykinin Receptor	*TACR1*	NK-1R C-term variant (2)	GR73632	CP96345	Mainly brain, GI tract, and lung	[70,71]
Opioid Receptors	*OPRD1*	DOR-1 6TM variant (4) 1TM variant (3)	DPDPE Enkephalin Deltorphin SNC-80	Naltrindole	Mainly brain	[72]
	*OPRK1*	KOR-1 N-term variant (5) 6TM variant (2) 1TM variant (1)	U50,488H U69,593 Nalfurafine Dynorphins	nor-BNI DIPPA JDTic	Mainly brain and prostate	[72]
	*OPRL1*	ORL-1/KOR-3 N-term variant (18) 6TM variant (3) 1TM variant (1)	SR-8993 Ro65-6570 N/OFQ	J-113,397 JTC-801	Mainly brain and blood	[72]
	*OPRM1*	MOR-1 C-term variant (12) N-term variant (1) 6TM variant (4) 1TM variant (4)	DAMGO Morphine Fentanyl Methadone Β-Endorphin	β-FNA Naloxone Naltrexone Cyprodime CTAP	Mainly brain	[11,12,73]
Prostaglandin E Receptors	*PTGER3*	EP3 C-term variant (6)	SC-46275 ONO-AE-248	DG-041	Mainly kidney, uterus, and stomach	[74]
Serotonin Receptors	*HTR2A*	5-HT2A 6-TM variant (1)	TCB-2	Volinanserin	Mainly brain	[75,76]
	*HTR2C*	5-HT2C C-term variant (1) 6-TM variant (1)	Loreaserin	Agomelatine	Mainly brain	[77,78]
	*HTR4*	5-HT4 C-term variant (9) EL variant (1)	Mosapride Prucalopride Velusetrag	SDZ 205–557 SB-207266 GR113808	Mainly intestine and brain	[79,80]
	*HTR6*	5-HT6 6-TM variant (1)	WAY-181187 EMD-386088 E-6801	Ro 04-6790 SB271046	Mainly brain	[81]
	*HTR7*	5-HT7 C-term variant (3)	AS-19 LP-12 E-55888	SB-269970 SB-258719 JNJ-18038683	Mainly brain, heart, and GI tract	[82,83,84,85]

Listed human GPCR splice variants were obtained from the indicated literature. C-Term: Full-length 7 transmembrane domains (TM) C-terminal splice variants; N-term: 5′ splice variant; 6TM variant: truncated 6TM variant; 1TM variant: single TM variant; IL variant: intracellular loop variant; EL variant: extracellular loop variant. THC: tetrahydrocannabinol; 2-AG: 2-arachidonylglycerol; ACEA: arachidonyl-2′-chloroethylamide; DPDPE: [D-Pen^2^,D-Pen^5^]Enkephalin; DAMGO: D-Ala^2^, N-MePhe^4^, Gly-ol]-enkephalin); N/OFQ: nociceptin/orphanin FQ; β-FNA: β-Funaltrexamine; TCB-2: (7R)-3-bromo-2, 5-dimethoxy-bicyclo[4.2.0]octa-1,3,5-trien-7-yl]methanamine; DIPPA: 2-(3,4-Dichlorophenyl)-N-methyl-N-[(1S)-1-(3-isothiocyanatophenyl)-2-(1-pyrrolidinyl)ethyl]acetamide hydrochloride; CTAP: D-Phe-Cys-Tyr-D-Trp-Arg-Thr-Pen-Thr-NH2; DG-041: (2E)-3-[1-[(2,4-Dichlorophenyl)methyl]-5-fluoro-3-methyl-1H-indol-7-yl]-N-[(4,5-dichloro-2-thienyl)sulfonyl]-2-propenamide; AS-19: (2S)-5-(1,3,5-Trimethylpyrazol-4-yl)-2-(dimethylamino)tetralin; LP-12: 4-(1,1′-Biphenyl)-2-yl-N-(1,2,3,4-tetrahydro-1-naphthalenyl)-1-piperazinehexanamide hydrochloride; VD60: 3,4,22-3-demethoxycarbonyl-3-hydroxylmethyl-4-deacetyl-vindoline 3,4-thionocarbonate; GI: gastrointestinal; DRG: dorsal root ganglion.

**Table 2 biomolecules-11-01525-t002:** Amino acid sequences of encoded by alternative exons downstream of exon 3 in 7TM C-terminal variants.

Species	Variant	Exons	Amino Acid Sequence Downstream of Exon3
Mouse	mMOR-1	4	LENLEAE*T*APLP
	mMOR-1A	3b	VCAF
	mMOR-1B1	5a	KIDLF
	mMOR-1B2	5ab	KLLMWRAMPTFKRHLAIMLSLDN
	mMOR-1B3	5abc	TSLTLQ
	mMOR-1B4	5abcd	AHQKPQECLKCRCLSLTILVICLHFQHQQFFIMIKKNVS
	mMOR-1B5	5abcde	CV
	mMOR-1C	7/8/9	PTLAVSVAQIFTGYP*S*P*T*HVEKPCK*S*CMDRGMRNLLPDDGPRQESGEGQLGR
	mMOR-1D	8/9	RNEEPSS
	mMOR-1E/Eii/Eiiii/Eiv	6/7/8/9	KKKLD*S*QRGCVQHPV
	mMOR-1F	10/6/7/8/9	APCACVPGANRGQTKASDLLDLELETVGSHQADAETNPGPYEGSKCAEPLAISLVPLY
	mMOR-1H/-1i/-1J	4	LENLEAETAPLP
	mMOR-1O	7ab	PTLAVSVAQIFTGYP*S*P*T*HVEKPCK*S*CMDR
	mMOR-1P	15	IMKFEAIYPKLSFKSWALKYFTFIREKKRN*T*KAGALPPLPTCHAGSPSQAHRGVAAWLLPLRHMGPSYPS
	mMOR-1V/-1Vii	18/6/7/8/9	KQEKTKTK*S*AWEIWEQKEHTLLLGETHLTIQHLS
	mMOR-1U	7/19/8/9	PTLAVSVAQIFTGYP*S*P*T*HVEKPCK*S*CMDSVDCYNRKQQTGSLRKNKKKKKRRKNKQNILEAGISRGMRNLLPDDGPRQESGEGQLGR
	mMOR-1W(D2)	18/6/7/8/9	LAFGCCNEHHDQR
	mMOR-1T(D2)	4	LENLEAE*T*APLP
Human	hMOR-1	4	LENLEAE*T*APLP
	hMOR-1A	3b	VRSL
	hMOR-1B1	5a	KIDLFQKS*S*LLNCE
	hMOR-1B2	5ab	RERRQK*S*DW
	hMOR-1B3	5abc	GPPAKFVADQLAG
	hMOR-1B4	5abcd	S
	hMOR-1B5	5abcde	VELNLDCHCENAKPWPLS*Y*NAG
	hMOR-1O	O	PPLAVSMAQIFTRY*S*PP*T*HREKTCNDYMKR
	hMOR-1X	X	CLPIPSLSCWALEHGCLVVYPGPLQGPLVRYDLPAILHSSCLRGNTAPSPSGGAFLLS
	hMOR-1Y	Y/5abc	IRDPISNLPRV*S*VF
	hMOR-1i	4	LENLEAE*T*APLP
Rat	rMOR-1	4	LENLEAE*T*APLP
	rMOR-1A	3b	VCAF
	rMOR-1B1	5a	KIDLF
	rMOR-1B2	5ab	EPQSVET
	rMOR-1C1	5abc	PALAVSVAQIFTGYP*S*P*T*HGEKPCK*S*YRDRPRPCGRTW*S*LK*S*RAE*S*NVENHFHCGAALIYNNVNFI
	rMOR-1C2	5abcd	PALAVSVAQIFTGYP*S*P*T*HGEKPCK*S*YRDRPRPCGRTW*S*LK*S*RAE*S*NVENHFHCGAALIYNNELKIGPVSWLQMPAHVLVRPW
	rMOR-1D	5abcde	T
	rMOR-1H1/H2/i1/i2/i3	4	LENLEAE*T*APLP
	rMOR-1P	15	GAEL

The amino acid sequences resulting from 3′ splicing are presented. S, T, and Y are predicted phosphorylation sites, labeled with red and italic letters, and highlighted with yellow. Underlined and yellow-highlighted sequences are predicted phosphorylation codes, PxPxxE/D or PxxPxxE/D, for β-arrestin binding based on crystal G protein coupled receptors (GPCR) structures [141]. Adapted from Pasternak and Pan 2013 [12].

**Table 3 biomolecules-11-01525-t003:** mRNA expression levels of all 7TM, 6TM, and 1TM splice variants in brain regions of four inbred mouse strains.

Strain	Brain Region	All 7TM Variants (mE1-2)		All 6TM Variants		All 1TM Variants	
		Expression (E^-ΔC(t)^)	% of all 7TM Variants	Expression (E^-ΔC(t)^)	% of All 7TM Variants	Expression (E^-ΔC(t)^)	% of All 7TM Variants
129P3	Pfc	0.012789	100	0.000284	3.0	0.000406	4.2
	Str	0.027871	100	0.001179	2.2	0.002181	4.0
	Tha	0.026575	100	0.001264	2.4	0.001814	3.4
	Hyp	0.036192	100	0.001170	2.6	0.002670	6.0
	Hip	0.021809	100	0.000873	3.2	0.001654	6.1
	PAG	0.020924	100	0.001090	2.4	0.001510	3.3
	BS	0.016948	100	0.001291	4.1	0.001646	5.2
	Cb	0.002093	100	0.000057	5.0	0.000175	15.5
	Spc	0.009637	100	0.001971	11.9	0.001239	7.4
	WB	0.008726	100	0.000647	13.5	0.001634	34.1
C57Bl/6J	Pfc	0.003384	100	0.000215	7.6	0.000206	7.3
	Str	0.008820	100	0.000327	3.2	0.000603	5.9
	Tha	0.029864	100	0.003211	8.0	0.002239	5.5
	Hyp	0.011862	100	0.000978	4.6	0.001090	5.1
	Hip	0.011638	100	0.000491	3.7	0.000720	5.4
	PAG	0.029910	100	0.013060	20.9	0.003317	5.3
	BS	0.012993	100	0.000670	3.9	0.001145	6.7
	Cb	0.003403	100	0.000078	6.2	0.000187	14.8
	Spc	0.009516	100	0.001087	7.5	0.001087	7.5
	WB	0.008982	100	0.000477	9.0	0.000416	7.8
SJL/J	Pfc	0.006287	100	0.000141	4.9	0.000375	13.0
	Str	0.011500	100	0.000568	6.7	0.000518	6.1
	Tha	0.022085	100	0.000645	2.4	0.001472	5.4
	Hyp	0.032938	100	0.002588	6.2	0.002166	5.2
	Hip	0.016593	100	0.000701	5.6	0.001084	8.7
	PAG	0.017955	100	0.000748	3.0	0.000869	3.5
	BS	0.029092	100	0.001523	4.4	0.001917	5.5
	Cb	0.003475	100	0.000244	17.7	0.000253	18.3
	Spc	0.030211	100	0.002908	5.9	0.002011	4.1
	WB	0.010318	100	0.000371	2.7	0.001296	9.3
SWR/J	Pfc	0.042361	100	0.000371	4.1	0.000681	7.5
	Str	0.012889	100	0.000384	2.2	0.001083	6.3
	Tha	0.010931	100	0.000280	1.5	0.000962	5.0
	Hyp	0.030392	100	0.001608	2.2	0.003108	4.2
	Hip	0.010829	100	0.000232	2.2	0.000661	6.3
	PAG	0.026677	100	0.000858	1.5	0.001792	3.2
	BS	0.020672	100	0.001450	4.0	0.001437	4.0
	Cb	0.014594	100	0.000598	9.8	0.000583	9.6
	Spc	0.031986	100	0.002458	2.8	0.003501	4.0
	WB	0.018681	100	0.000599	2.4	0.001033	4.2

The expression level (E^-ΔC(t)^) was calculated from Table S2 of Xu et al., 2014 [151]. The level of all 7TM variants was determined by qPCRs using mE1-2 primers. The level of all 6TM variants was calculated by adding the E^-ΔC(t^ values of individual five 6TM variants together, including mMOR-1G, mMOR-1K, mMOR-1L, mMOR-1M, and mMOR-1N. The level of all 1TM variants was calculated by adding the E^-ΔC(t^ values of individual five 1TM variants together, including mMOR-1Q, mMOR-1R, mMOR-1S, mMOR-1T, and mMOR-1Z. Percentage (%) of all 7TM variants was calculated by the formula: E^-ΔC(t^ value of 6TM or 1TM variants/ E^-ΔC(t^ value of mE1-2 × 100, so all 7TM variants are always as 100%. Pfc, prefrontal cortex; Str, striatum; Tha, thalamus; Hyp, hypothalamus; Hip, hippocampus; PAG, periaqueductal gray; BS, brain stem; Cb, cerebelum; Spc, spinal cord; WB, whole brain.

**Table 4 biomolecules-11-01525-t004:** *Oprm1* KO mouse models.

KO Mouse Model	Variant		
	7TM	6TM	1TM
E1 KO	Lost	Retained	Lost
E11 KO	Retained	Lost	Retained
E1/E11 KO	Lost	Lost	Lost
Triple KO (E1+DOR-1+KOR-1)	Lost	Retained	Lost

## Data Availability

Not applicable.

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
