# Peer review of "Alternative Pre-mRNA Splicing of the Mu Opioid Receptor Gene, OPRM1: Insight into Complex Mu Opioid Actions"

_biomolecules, 2021, doi:10.3390/biom11101525_

Round 1

Reviewer 1 Report

biomolecules-1381444

Alternative pre-mRNA splicing of the mu opioid receptor gene, OPRM1: insight into complex mu opioid actions

Shan Liu1, Wenjia Kang, Anna Abrimian , Jin Xu, Luca Cartegni, Susruta Majumdar, Patrick Hesketh, Alex Bekker and Ying-Xian Pan

This is a comprehensive review for a mu opioid receptor gene, OPRM1 especially focusing on splicing isoforms. The authors categorize isoforms into three major types based on the receptor structure, full-length 7TM C-terminal variants, truncated 6TM and single TM variants. They would like to draw a conclusion that the OPRM1 variants can be targets for development of novel opioid analgesics that are potent against multiple types of pain, at the same time they devoid of many side-effects associated with traditional opiates. It seems to me the authors provide an excellent overview of OPRM1 alternative splicing isoforms and their functional relevance in opioid pharmacology.

This is a wonderful review article to understand OPRM1 isoforms and functions. This review is quite interesting and useful to both basic scientists and clinicians.

I have a few suggestions to improve this manuscript as follows.

1) It would be wonderful if the authors could provide information for the ratio of 7TM, 6TM and 1TM mRNAs in brain or cells.

2) Although the authors describe mechanism and function of alternative splicing in one section, there is no information about mechanism for alternative splicing regulation in OPRM1. It would be nicer if the authors could offer regulatory cis-elements and trans-acting factors for OPRM1 alternative splicing.

Author Response

Reviewer 1

Alternative pre-mRNA splicing of the mu opioid receptor gene, OPRM1: insight into complex mu opioid actions

Shan Liu1, Wenjia Kang, Anna Abrimian , Jin Xu, Luca Cartegni, Susruta Majumdar, Patrick Hesketh, Alex Bekker and Ying-Xian Pan

This is a comprehensive review for a mu opioid receptor gene, OPRM1 especially focusing on splicing isoforms. The authors categorize isoforms into three major types based on the receptor structure, full-length 7TM C-terminal variants, truncated 6TM and single TM variants. They would like to draw a conclusion that the OPRM1 variants can be targets for development of novel opioid analgesics that are potent against multiple types of pain, at the same time they devoid of many side-effects associated with traditional opiates. It seems to me the authors provide an excellent overview of OPRM1 alternative splicing isoforms and their functional relevance in opioid pharmacology.

This is a wonderful review article to understand OPRM1 isoforms and functions. This review is quite interesting and useful to both basic scientists and clinicians.

Response: We thank for the reviewer’s constructive comments.

I have a few suggestions to improve this manuscript as follows.

  1. It would be wonderful if the authors could provide information for the ratio of 7TM, 6TM and 1TM mRNAs in brain or cells.

Response: The reviewer raised a legitimate issue regarding the expression ratio of 7TM, 6TM and 1TM at mRNA level. In our published qPCR data, we mainly quantified the levels for each individual splice variant among brain regions and mouse strains and did not calculate based on these three types. In the revised manuscript, we include a new table (Table 3) with quantification of each splice variant type based on the original qPCR data (Table S2) of our PLoS One 2014 paper [153] (PMID: 25343478). Since the qPCR data using mE1-2 primers represented the expression levels of all 7TM variants, we normalized the levels of all 6TM or all 1TM variants with all 7TM variants to obtain the percentage (%) of all 7TM variants. We also discuss the data in the main text.

  1. Although the authors describe mechanism and function of alternative splicing in one section, there is no information about mechanism for alternative splicing regulation in OPRM1. It would be nicer if the authors could offer regulatory cis-elements and trans-acting factors for OPRM1 alternative splicing.

Response: The reviewer made a good point. One of our research projects is to decipher the molecular mechanisms underlying OPRM1 alternative splicing including cis-acting elements and trans-acting factors by using multidisciplinary approaches including systematic deletion on minigene constructs, EMSA, RNA affinity purification coupled with proteomics, and siRNA thigh-throughput screening. Using some of these approaches, we revealed an intronic single nucleotide polymorphism (SNP)-containing cis-acting element located near the 3’ of exon 2 that can modulate OPRM1 alternative splicing through its interaction with hnRNPH, a splicing factor, and suggested a functional link between an SNP-containing cis-acting element or intronic splicing enhancer and the severity of heroin addiction. The results were published in our J Neurosci 2014 paper ([212] (PMID: 25122903). We now include Section 6 to discuss the data. In the past several years, we have conducted extensive studies on exploring the mechanisms of OPRM1 alternative splicing, which is the main goal of my R01 (DA042888). We expect to publish our findings in the near future.

Reviewer 2 Report

I was pleased to read an article by Dr. Shan Liu and colleagues for professional
review. The correspondent author of the work, Dr. Ying-Xian Pan,
was probably the closest associate of Prof. Gawril Pasternak,
the recently deceased emblematic figure in opioid research.
Based on my forty years of laboratory work on opioid peptides, opiates and
and their receptors, I dare say that this summary paper is suitable for publication.
The dissertation describes the single OPRM1 gene polymorphism, exon-intron distributions,
and the surprising multitude of splice variants in detail and with convincing thoroughness.
Functional relevance of the splice variants is mainly illustrated with the biochemical
and pharmacological properties of a recently developed IBNtxA synthetic alkaloid derivative.
3’-iodobenzoylnaltrexamide described to be antinociceptive (I think this term is better
than analgesic in animal experiments!), which is little bit surprising to me.
This compound has N-cyclopropylmethyl side chain carrying strong antagonist properties
in a number of biochemical, e.g. receptor binding assays, as well as pharmacological test.
This point should be discussed.
Although I am not a native English speaker, I still think that
the ‘spiced’ spelling of the blue boxes shown in several figures is correctly ‘spliced’.
It is somewhat unusual for me to take over the large number of figures
described in previous publications. What justified this?
If new data has been added to the figures, this should also be indicated.

Author Response

Reviewer 2

I was pleased to read an article by Dr. Shan Liu and colleagues for professional review. The orrespondent author of the work, Dr. Ying-Xian Pan, was probably the closest associate of Prof. Gawril Pasternak,
the recently deceased emblematic figure in opioid research. Based on my forty years of laboratory work on opioid peptides, opiates and and their receptors, I dare say that this summary paper is suitable for publication. The dissertation describes the single OPRM1 gene polymorphism, exon-intron distributions, and the surprising multitude of splice variants in detail and with convincing thoroughness

Response: Thanks for the reviewer’s positive comments.

Functional relevance of the splice variants is mainly illustrated with the biochemical and pharmacological properties of a recently developed IBNtxA synthetic alkaloid derivative. 3’-iodobenzoylnaltrexamide described to be antinociceptive (I think this term is better than analgesic in animal experiments!), which is little bit surprising to me.

Response: Opioid analgesic or analgesia is commonly used in opioid field and we have used in almost all our publications and presentations in the last three decades. However, antinociceptive could also be used with similar meaning to analgesia.

This compound has N-cyclopropylmethyl side chain carrying strong antagonist properties in a number of biochemical, e.g. receptor binding assays, as well as pharmacological test. This point should be discussed. Although I am not a native English speaker, I still think that the ‘spiced’ spelling of the blue boxes shown in several figures is correctly ‘spliced’.

Response: The reviewer makes a good point. The N-cyclopropyl group on the morphinan template is indeed believed to be an antagonist pharmacophore, an example being naltrexone. However, it is now well understood that incorporating an aryl amido group at the 6-position of the morphinan template leads to agonism at the 6TM site in IBNtxA [198] (PMID: 22106286). Similarly, such an incorporation in MP1104 confers its characterization as a kappa agonist at KOR-1 site [200] (PMID:29307491), while MP1207 and MP1208 with the incorporation become mixed mu/kappa agonists [185] (PMID: 33555255). Therefore, the 6-position arylamido group on the morphinan template puts the receptor in an agonist conformation. We discussed this in main text. We also checked “spliced” spelling throughout figures.

It is somewhat unusual for me to take over the large number of figures described in previous publications. What justified this? If new data has been added to the figures, this should also be indicated.

Response: In revised manuscript, we deleted 6 figures and added one new Table (Table 3) based on the comment of the reviewer 1.